# Myelin Oligodendrocyte Glycoprotein Antibody-Associated Disease: Pathophysiology, Clinical Patterns, and Therapeutic Challenges of Intractable and Severe Forms

**DOI:** 10.3390/ijms26178538

**Published:** 2025-09-02

**Authors:** Tatsuro Misu

**Affiliations:** Department of Neurology, Tohoku University Hospital, 1-1 Seiryomachi, Aobaku, Sendai 980-8574, Japan; tatsuro.misu.d8@tohoku.ac.jp; Tel.: +81-22-717-7189

**Keywords:** myelin oligodendrocyte glycoprotein, demyelination, biologics, complement, autoimmunity

## Abstract

Myelin oligodendrocyte glycoprotein (MOG) antibody-associated disease (MOGAD) is characterized by the predominance of optic neuritis, myelitis, acute disseminated encephalomyelitis (ADEM), and cortical encephalitis, and can be diagnosed by the presence of pathogenic immunoglobulin G (IgG) antibodies targeting the extracellular domain of MOG in the serum and cerebrospinal fluid (CSF). Initially considered a variant of multiple sclerosis (MS) or neuromyelitis optica spectrum disorder (NMOSD), it is now widely recognized as a separate entity, supported by converging evidence from serological, pathological, and clinical studies. Patients with MOGAD often exhibit better recovery from acute attacks; however, their clinical and pathological features vary based on the immunological role of MOG-IgG via antibody- or complement-mediated perivenous demyelinating pathology, in addition to MOG-specific cellular immunity, resulting in heterogeneous demyelinated lesions from vanishing benign forms to tissue necrosis, even though MOGAD is not a mild disease. The key is the immunological mechanism of devastating lesion coalescence and long-term degenerating mechanisms, which may still accrue, particularly in the relapsing, progressing, and aggressive clinical course of encephalomyelitis. The warning features of the severe clinical forms are: (1) fulminant acute multifocal lesions or multiphasic ADEM transitioning to diffuse (Schilder-type) or tumefactive lesions; (2) cortical or subcortical lesions related to brain atrophy and/or refractory epilepsy (Rasmussen-type); (3) longitudinally extended spinal cord lesions severely affected with residual symptoms. In addition, it is cautious for patients refractory to acute stage early 1st treatment including intravenous methylprednisolone treatment and apheresis with residual symptoms and relapse activity with immunoglobulin and other 2nd line treatments including B cell depletion therapy. Persistent MOG-IgG high titration, intrathecal production of MOG-IgG, and suggestive markers of higher disease activity, such as cerebrospinal fluid interleukin-6 and complement C5b-9, could be identified as promising markers of higher disease activity, worsening of disability, and poor prognosis, and used to identify signs of escalating treatment strategies. It is promising of currently ongoing investigational antibodies against anti-interleukin-6 receptor and the neonatal Fc receptor. Moreover, due to possible refractory issues such as the intrathecal production of autoantibody and the involvement of complement in the worsening of the lesion, further developments of other mechanisms of action such as chimeric antigen receptor T-cell (CAR-T) and anti-complement therapies are warranted in the future.

## 1. Introduction

Myelin oligodendrocyte glycoprotein antibody-associated disease (MOGAD) is a distinct demyelinating disorder of the central nervous system (CNS) characterized by the presence of pathogenic immunoglobulin G (IgG) antibodies targeting the extracellular domain of myelin oligodendrocyte glycoprotein (MOG), a glycoprotein expressed on the surface of oligodendrocytes and the outermost lamellae of CNS myelin sheaths. Initially considered a variant of multiple sclerosis (MS) or neuromyelitis optica spectrum disorder (NMOSD), MOGAD is now widely recognized as a separate nosological entity, supported by converging evidence from serological, pathological, and clinical studies [1,2].

Clinically, MOGAD manifests as diverse phenotypes, including optic neuritis (ON), transverse myelitis (TM), acute disseminated encephalomyelitis (ADEM), and brainstem encephalitis. Notably, disease presentation can vary by age group; children more commonly exhibit ADEM-like symptoms, whereas adults typically present with ON or TM [3,4]. These attacks are often monophasic, but may recur, with some patients experiencing frequent and disabling relapses. Unlike patients with AQP4-IgG^+^ NMOSD, those with MOGAD often exhibit better recovery from acute attacks. However, long-term disabilities may still occur, particularly in relapsing forms.

Serologically, MOG-IgG is most reliably detected using cell-based assays that use full-length human MOG as the target antigen. These antibodies are predominantly of the IgG1 subtype and thought to be pathogenic via complement- and antibody-dependent cellular cytotoxicity [5,6]. However, immunopathological mechanisms underlying MOGAD remain unclear. Autopsy and biopsy findings reveal perivenous demyelination, axon preservation, and a relative lack of astrocyte loss, contrasting with AQP4-NMOSD [7,8]. These findings, along with distinct cerebrospinal fluid (CSF) profiles and magnetic resonance imaging (MRI) characteristics, reinforce the nosological distinction between MS and NMOSD.

Epidemiological studies have suggested that MOGAD accounts for a significant proportion of acquired demyelinating syndromes, particularly in children with ADEM/encephalitis and adults with AQP4-IgG seronegative NMOSD [9,10]. The annual incidence is approximately 1.6–3.4 per million, although the true prevalence is likely underestimated owing to diagnostic challenges and evolving serological testing standards [5,11]. The MOGAD burden is compounded by its unpredictability; although many patients respond well to steroids and immunotherapies, others experience recurrent or severe attacks that are difficult to control [5,12].

Owing to this clinical heterogeneity, growing attention has been paid to the severe and treatment-refractory subtypes of MOGAD. These forms may be associated with persistent or relapsing disease, poor steroid responsiveness, and atypical imaging patterns. A deeper understanding of immunopathological mechanisms, including the role of intrathecal MOG-IgG synthesis and complement activation, is essential for improving diagnostic precision and tailoring therapy. This review aims to summarize the current knowledge on the pathophysiological basis, imaging characteristics, immunological biomarkers, and therapeutic challenges of refractory and severe MOGAD, to guide future clinical and translational research.

## 2. Pathological Insights

### 2.1. Pathological Features of MOGAD

MOGAD is pathologically distinct from MS and aquaporin-4 (AQP4)-IgG^+^ NMOSD and exhibits characteristic features that reflect its unique immunopathogenesis [13]. Perivenous demyelination and the dominance of CD4^+^ T-cell infiltration [7,14], along with granulocytes such as neutrophils and eosinophils, often reminiscent of monophasic ADEM, are considered hallmarks of MOGAD in the acute stage [15]. These lesions typically originate around small venules, and complement deposition is relatively rare in MOG antibody^+^ ADEM cases [7] compared to NMOSD cases; however, some ADEM-like cases resemble type II MS pathology [16]. Unlike the sharply demarcated confluent plaques seen in prototypic MS, MOGAD lesions tend to be ill-defined and patchy, with relative preservation of axons and oligodendrocytes, particularly early lesions [7,8]. This contrasts with the pathology in MS, where CD8^+^ T cells are predominant and chronic microglial activation occurs in chronically expanding lesion limbs, and in AQP4-NMOSD, which shows marked astrocyte loss and complement-mediated tissue necrosis [17]. These histopathological differences are associated with favorable clinical recovery in patients with MOGAD; clinically relevant T2-lesions in MOGAD resolve more completely and frequently than those in MS and NMOSD [18].

### 2.2. Smoldering Confluent MS-like Lesions Are Rare in MOGAD

An important aspect of MOGAD pathology is the rapid centrifugal expansion of demyelinating lesions from the perivenous epicenter, which is driven by T cell infiltration and secondary activation of innate immune cells. Several multiple perivenous demyelinating lesions often coalesce and grow into larger lesions [7,8]. This radial lesion growth resembles the “smoldering” confluent lesions described in progressive MS, but the progression scale is different and lacks the chronic microglial rim that typifies such MS plaques [13]. Moreover, although MS is associated with oligodendrocyte apoptosis and axonal transection, MOGAD exhibits limited oligodendrocyte damage, potentially from antibody-mediated myelin disruption, without direct cytotoxicity to myelin-producing cells [8,13].

### 2.3. Pathogenesis of MOG-IgG

#### 2.3.1. Complement-Mediated

The isotypes of autoantibodies against MOG and AQP4 are mainly IgG1 can mediate complement-dependent cytotoxicity (CDC) [19] in addition to antibody-mediated cytotoxicity (ADCC) [20]. The assembly of AQP4 into orthogonal arrays of particles (OAPs), consistent with tetrameric AQP4-M23 components, is required for complement activation, which is influenced by the interaction of the Fc portion of AQP4-IgG with the OAP structure [21]. Binding of C1q to the Fc portion of AQP4-IgG in astrocytes activates C1r and C1s, which then cleave C4 and C2. These complement proteins are also cleaved by lectin-associated serine proteases [22], which are activated when mannose-binding lectins encounter conserved carbohydrate motifs in pathogens (lectin pathways). The cleaved products C4 and C2 then combine to form C3 convertase, which further cleaves C3 into C3b and C3a. C3b then associates with C4bC2a to form C5 convertase, finally activating cascades of C5b, C6, C7, C8, and C9 for the membrane attack complex (MAC) [22]. This pathway has been observed in clinical and experimental NMOSD pathology [17,23], and MAC biomarkers are associated with disease activity [24,25]. In addition, the alternative pathway is activated when C3 spontaneously hydrolyzes to form a C3 convertase, C3(H_2_O)Bb, in the presence of factors B and D, leading to additional C3 cleavage and the eventual formation of C3 and C5 convertases [22]. This pathway is also involved in the MAC formation. These three pathways work in concert to protect the host from pathogenic invasion by activated components, such as C3aR or C5aR, including activated macrophage-induced phagocytosis; anaphylatoxins, including C3a and C5a, inducing neutrophil recruitment; and activation of innate and adaptive immunity, including B and T cells [22], some of which are reported to be strongly or moderately associated with NMOSD pathology [21,26,27]. AQP4-IgG showed markedly higher levels of CDC than MOG-IgG in vitro [19] together with marked deposition of MAC in pathological studies [28], which could be due to the difference in C1q binding to targeted molecules between AQP4-IgG and MOG-IgG, initiating a classical complement pathway [29]. In contrast, C1q is histologically observed both in the biopsied brain tissue of pediatric patients with MOGAD and MOG-IgG-induced macaque experimental autoimmune encephalomyelitis (EAE) [30], suggesting that CDC in MOGAD is relatively weaker than in NMOSD but has a common mechanism.

Interestingly, although complement deposition is not always prominent in MOGAD and is stage-dependent [7,8], some necrotizing lesions exhibit C9neo deposition, indicating complement activation and MAC formation [8,31]. An in vitro study suggested that MOG-IgG-induced complement-mediated cytotoxicity [32] and MOGAD encompasses a spectrum of several histological phenotypes, including benign and malignant. Autopsy and biopsy revealed widespread confluent demyelination, perivascular necrosis, and complement-mediated cytotoxicity, akin to pattern II MS lesions, in a subset of patients, particularly those with refractory disease [33,34]. An autopsy also showed diffuse multi-coalesced demyelinated lesions with deep complement deposition in a patient with progressive extension of diffuse and multifocal white matter lesions [8]. These cases may underlie the severe clinical presentation and reduced responsiveness to immunotherapy, although the definitive biomarkers for predicting such pathologies remain unclear.

#### 2.3.2. Antibody-Mediated

In an in vitro study, ADCC induces the striking loss of the thin filaments and microtubule cytoskeleton in cultured oligodendrocytes [35]. In an in vivo microinjection model, MOG-IgG itself without complement can cause myelin changes and altered expression of axonal proteins without marked inflammation and other tissue damage and could recover well within 2 weeks [36]. Moreover, in an EAE model, MOG-IgG-induced demyelination is equally mediated by CDC and ADCC, suggesting a diverse mechanism of pathological characteristics involved in MOGAD, with or without complement-induced demyelination and tissue necrosis [37]. Moreover, its ADCC can be activated by FcγR activation [37], which may also be essential for cognate T cell activation via antigen-presenting cells.

#### 2.3.3. Cellular Immunity, Including MOG-Specific and Innate Immunity

Another aspect of MOG-IgG cytotoxicity is antibody-dependent cellular phagocytosis (ADCP), which could be identified by its functional role on in vitro MOG-expressing cells [20]. MOG-IgG includes not only IgG1 but also IgG2, IgG3, and IgG4, all of which could induce ADCP; in contrast, CDC could be observed by IgG1 and IgG3, suggesting that patient-derived antibodies must have multiple mechanisms of cytotoxic effects against targeted myelin sheaths [20], possibly linked to a unique accumulation of macrophages in perivenous demyelinating lesions in the ADEM phenotype of MOGAD [7]. In EAE, MOG-specific CD4+ and CD8+ T cells are required for the induction of EAE in B cell-independent conditions, which can explain the suboptimal responses to anti-CD20 therapy in patients with MOGAD compared with NMOSD [38]. Several epitopes of MOG-specific T cells have been reported in patients with MOGAD, including extracellular and intracellular epitopes [39,40], consistent with various epitopes of MOG-IgG [31], suggesting various encephalitogenic roles and possible epitope spreading. Pathogenic patient-derived MOG-IgG enhanced cognate MOG-specific T cell and macrophage infiltration in experimental models, which could drive tissue inflammation and demyelination via an ensemble of various T and B cell epitope spreading and disease worsening [31]. Considering its longitudinal history, there is no doubt that MOG-associated disease has been most deeply studied regarding several points of the above-mentioned pathological mechanisms, some of which suggest upstream factors related to the worsening of MOGAD.

## 3. Clinical Features of MOGAD

The clinical manifestations of MOGAD are heterogeneous, ranging from ON [41], myelitis [42], tumefactive disease [43], multifocal ADEM, or cortical encephalitis [44,45]. However, isolated ON was the most frequent clinical presentation (approximately 50%) in both children and adults [46], and the first episode of ON reduced the risk of expanded disability scale score (EDSS) progression compared to myelitis [47]. Two-thirds of the patients had prodromal symptoms, such as rhinorrhea, sore throat, low-grade fever, or cough, owing to a presumed or confirmed infection [48], which is often encountered in pediatric patients and patients after several kinds of vaccinations, including against SARS-CoV-2. ADEM was more frequent and associated with a better prognosis in children (36.7%) than in adults (5.6%) [46]. In contrast, relapse in adults, particularly those with manifestations of ON and myelitis [49], is relatively higher than that in children [46], with the probabilities of reaching a first relapse after 2 and 5 years being 44.8% and 61.8%, respectively [4]. At 2 years, monophasic ADEM often becomes MOG-IgG^−^ (64.2%); in contrast, most relapsing cases have persistent MOG-IgG [46]. Therefore, MOGAD is generally considered better than NMOSD, even though it is not a mild disease [43]. Therefore, it is important to pay attention to clinical features of various benign and malignant factors that may influence clinical outcomes, including disease relapse activity and worsening of disability (Table 1).

### 3.1. Red Flags for Severe MOGAD

Severe MOGAD often presents with bilateral ON, MOG-associated encephalitis, cortical encephalitis with seizures, and longitudinally extensive transverse myelitis (LETM) resulting in persistent motor deficits [50,51]. Moderate-to-poor recovery after acute immunotherapy is frequently observed in patients with LETM (40%), bilateral ON (32%), ADEM (17%), short myelitis (11%), and unilateral ON (11%), with a relatively higher ratio in the onset group [42] (Table 1). In addition, current smoking status influences the onset attack recovery of patients with MOGAD and has a higher risk of disability from the onset attack and first optic neuritis attack than never smokers (odds ratio 2.9) [52].

#### 3.1.1. Transverse Myelitis

Incomplete recovery from paraplegia and bladder dysfunction may necessitate aggressive interventions [53]. Myelitis attacks in acute MOGAD are often severe, with transverse symptoms including motor weakness (83%), numbness (89%), sphincter symptoms (83%), cane (41%), or wheelchair (33%), often with longitudinally extended lesions over three vertebral segments (79%), and are often categorized as EDSS 6.0 [48]. In addition, neurological findings in the acute stage involve both spasticity/hyperreflexia and flaccid areflexia due to LETM, some of which meet the criteria for acute flaccid myelitis with preceding prodromal infectious symptoms, and MRI findings of the central gray matter, which could be of prognostic value [54]. In particular, neurogenic bladder results from the frequent involvement of conus lesions on MRI, resulting in more than 60% residual autonomic dysfunction in MOGAD than in NMOSD [48,55]; some of these patients require long-term catheterization (20–44%). In addition, 6–7% of patients needed a gait aid (EDSS > 6.0) at the last follow-up [48,55]. In the chronic stage, patients have progressive myelopathy with periventricular lesions and a relatively low titer of MOG-IgG, who are refractory to B cell depletion but treated with monthly intravenous immunoglobulin (IVIG) [56]. In addition to the whole-brain fraction, upper cervical cord atrophy (UCCA) is associated with the disability score of patient-derived disease steps in MOGAD [57].

#### 3.1.2. Optic Neuritis

Poor visual recovery with recurrence, despite high-dose steroids or progression to bilateral involvement, often signifies refractory disease. Maintenance therapy with intravenous immunoglobulin (IVIG), rituximab, or tocilizumab has shown variable success [58]. Acute symptoms of ON are often severe, with vision loss of less than 20/200, often bilateral (40%), papillary edema (80%), and eyeball pain (90%); however, corticosteroid treatment is relatively effective [42], particularly in milder forms than AQP4-IgG^+^ NMOSD and double-negative NMOSD [47]. However, 12 of the 75 patients experienced moderate-to-severe permanent visual dysfunction [42]. Poor (6.2%) and incomplete (19.4%) visual acuity recovery were previously observed with poor visual field abnormalities (16.9%); however, the lack of or milder optic nerve sheath enhancement was associated with incomplete recovery, which could be a prognostic factor for MOGAD [59]. Optical coherence tomography (OCT) has suggested that peripapillary retinal nerve fiber layer (pRNFL) thickness measured acutely frequently demonstrates swelling, and its thickness in acute optic neuritis can differentiate MOGAD from MS [60]. In contrast, a negative correlation between follow-up pRNFL thickness and the latest follow-up visual acuity was noted, which suggested a degenerating mechanism in MOGAD [61].

#### 3.1.3. Encephalopathy and Seizures

In children and young adults, ADEM and MOG-associated encephalitis may present with monophasic or multiphasic ADEM with cerebral symptoms of encephalopathy, including decreased levels of consciousness (100%) and seizures (30–60%) [43,62]. Most monophasic ADEM cases are treated with first-line immunotherapy, resulting in symptom and MRI lesion resolution [63]. In contrast, in a pediatric ADEM cohort (*n* = 46), modified Ranking Scale (mRS) > 2 was observed in six cases (13%) at the last follow-up, eight were multiphasic ADEM (MDEM), and five were ADEM followed by relapses in ON or myelitis. Half of the MDEM cases resulted in severe disability (mRS 4 and 2 in two cases each), suggesting MDEM as a red flag [64]. In addition, persistent epilepsy or cognitive impairment may occur, particularly in patients with extensive cortical involvement of leukodystrophy-like or predominant cortical involvement [43,45], some of whom have persistent brain atrophy and severe cognitive impairment [43,65]. Other refractory cases include acute symptomatic seizures and new-onset refractory status epilepticus (NORSE) owing to MOG-IgG-induced ADEM or hemispheric cortical encephalitis, which several lines of immunotherapy and anti-epileptic treatments cannot subside [66,67], resulting in refractory seizure-induced hemispheric brain atrophy, such as Rasmussen’s encephalitis. Most seizures that subsided with immunotherapy have focal patterns owing to a unilateral type of cortical encephalitis called FLAIR hyperintense lesions in anti-MOG-associated encephalitis with seizures (FLAMES) [44,68], and a combination of immunotherapy and anti-epileptic drugs is often required for long-term remission.

#### 3.1.4. Cognitive Impairment

Cognitive impairment in MOGAD is mostly a sequela of disease onset or relapsed symptoms, but it might be underestimated a progressive deterioration of cognition in MOGAD as observed in smoldering MS lesions [69]. In an adult cohort including 32 MOGAD cases (median age 29.4 years), several types of cognitive impairments were observed, such as mental flexibility (16.7%), attention (11.1~14.8%), and verbal working memory (10.3%) with reduced volumes of cerebral white matter and gray matter compared with controls, which is associated with a history of ADEM/ADEM-like episodes [70]. In contrast, in a pediatric MOGAD cohort (*n* = 109), in a median follow-up of 1.6 years, 15 of 82 cases and 14 of 59 cases with brain lesions had learning difficulties, probably because of reduced brain growth observed in 86 of 109 (79%) patients in acute stage, and 50 of 109 (46%) had brain atrophy at follow-up driven progressively by brain lesions [71].

### 3.2. Clinical Indicators of Refractory Activity

Although most cases of MOGAD respond favorably to corticosteroids and immunotherapy, a subset of patients experience severe or refractory disease, defined by criteria such as poor functional recovery; early relapse during sequential acute immunotherapy, including IVIG and plasma exchange (PLEX); and resistance to corticosteroids and immunosuppressants such as azathioprine, tacrolimus, mycophenolate mofetil, or other maintenance therapies [46,50,51]. Moreover, ongoing disease activity during maintenance immunotherapy with rituximab and IVIG is considered the motivation for tocilizumab trials [58,72]. In addition to these aggressive treatments, one case report attempted hematopoietic stem cell transplantation [73] resulting in a release from disease activity. Complications of other autoimmune diseases and autoimmune encephalitis are influenced by refractoriness to specific treatments of MOGAD [74,75]. The signs of refractory MOGAD are as follows:One or more relapse activities per 1–2 years [51,62].Relapsing trends at a dose of prednisolone <10 mg daily within 2 months of onset [51].Annualized relapse rate over 0.5~1.0 under immunotherapy [42,62].Ineffective first-line immunotherapy in the acute stage within 2–4 weeks [62].Progressive or relapsing symptoms and lesions treated with second-line immunotherapy [42,62,72].Residual symptoms under optimal treatments at the first onset [42].Symptoms of seizure and consciousness disturbance in encephalitis [67].Other complications of autoimmune diseases and autoimmune encephalitis [74,75,76], especially overlapping syndrome of anti-NMDAR encephalitis with MOGAD often require intensified care for status epilepsy especially in pediatric cases [77].

### 3.3. Pediatric vs. Adult Presentations

Children with MOGAD tend to have a high frequency of ADEM-like episodes which could influence on the increased ratio of monophasic course and better recovery comparted with adults; however; severe leukodystrophy-like phenotypes with diffuse white matter changes and developmental delays have also been reported [62,78]. In contrast; adults are more prone to ON-dominant forms. However; patients with recurrent brainstem or MOG encephalitis may experience significant residual disability [46]. The EDSS at the onset nadir over 6.0 is higher in pediatric patients (35.7%) than in adult patients (26.7%). In contrast; the EDSS over 3.0 at the last follow-up was higher in adults (20%) than in children (6.3%), suggesting a relatively good response to immunotherapy in pediatric patients [46]. However; the prognoses of major features; including ON and myelitis; are similar in several cohorts of children and adults. Interestingly; the rate of ADEM encephalopathy or brain lesions is approximately 50% in children [9] but less than 10% in adults. Severe and refractory cases due to brain lesions have also been observed in pediatric patients [66,79]. In a cohort of pediatric patients with MOGAD; early immunotherapy less than 1 week from onset was predictive of a monophasic course. In contrast; MOGAD with a relapsing course had a higher proportion of final EDSSs over 1 and 2 [80] and should not be handled with a good prognosis. Moreover; it is reported pediatric-onset patients had a higher likelihood of developing chronic epilepsy over 3-fold higher compared with adults and there is a age-related difference in cortical lesions distribution in MOGAD: parietal lobe involvement predominated in adults; whereas temporal lobe lesions were more frequently involved in children; potentially increasing vulnerability to seizures in pediatric-onset patients and should pay attention to status epilepsy needs multidisciplinary treatments [77].

**Table 1 ijms-26-08538-t001:** Candidates for relapsing and severe disability prognostic factors in MOGAD.

	Benign Factors	Malignant Factors	R/D	Refs.
Clinical features				
Onset age	Younger less than 10 years	Older age in adults	R/D	[9,46]
	(monophasic > multiphasic)	(relapse and poor recovery)		
Clinical course	Monophasic	Multiphasic/relapsing	R/-	[46,62]
	(EDSS < 1 in 79%)	(EDSS > 2 in 21%)	-/D	[80]
Onset of symptoms	Optic neuritis	Transverse myelitis	-/D	[42,81]
	(full recovery rate > 0.6)	(incomplete recovery rate > 0.7)		
	ADEM in children	Multiphasic DEM in children	R/-	[43,80]
	(mRS mostly within 1)	(mRS 2 to 4, EDSS 3~8)	-/D	[43,78]
Epilepsy	Well-controlled	NORSE/refractory or status epilepsy	-/D	[66,67]
Cognition	None	Learning difficulty in children (14~25%)	-/D	[71]
		(about 20%, particularly in 50% MDEM)	-/D	[62]
Cigarette smoking	None	Poor recovery from disability	-/D	[52]
Therapeutics (acute)	Markedly improved in 1st IVMP	Refractory to 1st and 2nd IVMP	-/D	[62,82]
	(EDSS at f/u < 1.0)	(EDSS at f/u > 2.0)		
	Good recovery after onset (EDSS 0)	Residual symptoms after treatment	-/D	[62]
	IVMP within 1 week (EDSS < 1.0)	Delayed IVMP (EDSS > 2.0)	R/D	[80]
	Over 3 months of immunotherapy	Early tapering of GC within 2 months	R/-	[42,51]
	Early apheresis/concomitant DMT use	non-apheresis/non-DMT	R/D	[83]
	(complete remission associated with DMT use (odds ratio 1.477), EDSS difference 0.0 vs. 2.5)			
Therapeutics (chronic)	No relapse in maintenance (6 months)	Disease activity with 2nd line treatment	R/D	[62,83]
	(ARR > 1.5 point reduction in RTX and IVIG)		R/-	[62]
	(EDSS 1.0 point reduction in IVIG)		-/D	[62]
**Biomarkers**				
MOG-IgG titer	Seronegative conversion	High in remission titration	R/-	[84]
	(95% relapse risk reduction)	(especially in titers > 1:2560 for relapsing)		
Intrathecal MOG-IgG	Isolated serum for optic neuritis	CSF persistent for severe phenotypes	-/D	[85,86]
	(EDSS < 3.0. in 82%)	(EDSS > 3.0. in 29%, >6.0. in 19%)	-/D	[87]
MRI	Vanishing lesions in initial treatment	Residual lesions after treatment	-/D	[18]
	No atrophy	Progressive atrophy	-/D	[43,88]
	Isolated multiple lesions	Leukodystrophy-like diffuse lesions	-/D	[69,78]
		(Transitional/Schilder type)		
Nf-L/Tau	Low in serum	High in acute stage with relapse/seizure	R/D	[89,90]
IL-6	Low in serum and CSF	High in acute stage severely disabled	R/D	[91,92]
C5b-9	Low in serum and CSF	High in patients with EDSS > 3.0.	-/D	[27,93]

Factors possibly linked to benign and malignant courses are listed, focusing on relapse (R) and disability (D) with related references. EDSS, Expanded Disability Status Scale; mRS, modified Rankin scale; NORSE, new-onset refractory status epilepticus; ADEM/MDEM, acute or multiphasic disseminated encephalitis; IVMP, intravenous methylprednisolone; GC, glucocorticoid; ARR, annualized relapse rate; RTX, rituximab; IVIG, intravenous immunoglobulin; CSF, cerebrospinal fluid; Nf-L, neurofilament L.

## 4. MRI and Other Biomarkers Predicting Severity

### 4.1. Typical MRI Features

MOGAD exhibits distinct MRI patterns that are essential for differentiating it from MS and AQP4^+^ NMOSD. Although these conditions often overlap in clinical presentation, they can be distinguished through careful evaluation of the lesion distribution, morphology, and enhancement patterns. Unlike MS, multiple isolated lesions are often observed; however, juxtacortical U-fiber sparing and the absence of Dawson’s fingers are typical features (Figure 1A). These lesions tend to involve the deep gray nuclei, brainstem, and supratentorial white matter, often symmetrically [94,95] and completely subside in benign cases (Figure 1B). MOGAD frequently presents as multifocal small-to-large lesions and poorly demarcated bilateral ADEM-like lesions in the brain (Figure 1C,D), particularly in children. Some of these lesions are focal or multifocal tumefactive lesions (Figure 1E). A hallmark of cortical presentation is FLAMES, showing unilateral [44] (Figure 1F) or bilateral medial [45] (Figure 1G) cortical swelling, often associated with seizures and a relatively benign prognosis [68]; however, there are severe cases refractory to inflammatory and anti-epileptic treatments [66,67]. Spinal cord lesions in MOGAD are usually longitudinally extensive and centrally located, frequently involving the conus medullaris and the thoracic segments. MOGAD lesions often had the gray matter “H sign” [53,96]. Brainstem and cerebellar peduncular lesions are more frequent in MOGAD than in NMOSD, with lesions often appearing in the middle cerebellar peduncles [4,97,98]. In contrast, area postrema lesions are relatively rare compared in NMOSD [4,28,99]. ON in MOGAD tends to be bilateral, with anterior optic nerve and optic disk involvement often associated with disk edema [41]. Orbital fat inflammation and perineural enhancement are more common than those in MS or AQP4-NMOSD [100], and optic chiasm involvement, when possible, is typically partial [95,101]. Importantly, MRI misinterpretation can lead to misdiagnosis of MOGAD as MS, particularly when corpus callosum or periventricular lesions are present. However, MOGAD typically lacks the chronic black holes, callosal thinning, central vein sign, and progressive diffuse atrophy seen in MS [94,101].

### 4.2. MRI Features Predicting Severe and Refractory MOGAD

Although MOGAD is generally monophasic and radiologically responsive to treatment, such as vanishing tumors, various atypical MRI features may correlate with treatment-resistant or severe phenotypes. These include:One or more relapse activities per recent 1 to 2 years [61]. Multifocal demyelinating lesions acutely transitioned to diffuse Schilder-like sclerosis [43] (Figure 1C,D).Leukodystrophy-like patterns with diffuse symmetric white matter changes [43,78,79] (Figure 1D).Tumefactive lesions often mimic gliomas or acute MS [4,73,102] (Figure 1E).Cortical lesions (Figure 1F) and atrophy (Figure 1G) with epilepsy, such as Rasmussen encephalitis (Figure H) [66,76].Extensive brainstem and cortical involvement are associated with worsening of disability [4,9], and concomitant brainstem lesions are associated with a higher mean EDSS (>3.0) during recovery [55].Longitudinally extended spinal cord lesions [42,53,103] and conus lesions are associated with autonomic failure [42,55].Bilateral redundant swollen optic nerves in the acute stage are often observed with relatively severe visual acuity loss in the nadir [41,103], of which a lack of or milder enhancement is a sign for poor prognosis [59].

Quantitative imaging has demonstrated the relative preservation of brain and spinal cord volumes in MOGAD compared to MS and NMOSD, supporting its general non-degenerative course [102] in MOGAD. However, repeated attacks and residual lesions can result in cumulative tissue damage causing atrophy, particularly in the spinal cord [53,96]. In addition, progressive atrophy in MOGAD can be observed in patients with refractory encephalitis, possibly with other autoimmune diseases such as NMDAR encephalitis [67,76] and refractory status epilepsy related to Rasmussen’s encephalitis [104]. Accurate recognition of these features is critical not only for diagnosis but also for identifying patients at risk for severe disease.

### 4.3. Biomarkers Predicting Severe and Refractory MOGAD

The identification of biomarkers to guide treatment decisions in MOGAD is an evolving field, especially the titration of MOG-IgG in serum and CSF, inflammatory cytokines, and tissue damage markers [105]. The candidates are as follows:Persistently high MOG-IgG titers during or after treatment are correlated with the risk of relapse and poor steroid response and require long-term immunosuppression [4,12,84,100,106].Serum and CSF cytokine and chemokine profiles, such as elevated interleukin-6 (IL-6) [92,107,108], IL-8 [107,108], and B cell activating factor (BAFF) [108] levels, reflect active B cell-mediated inflammation and predict the need for aggressive therapy [91,92,107,108] associated with disease severity in general, of which BAFF levels predict a lower risk of relapse [93].Intrathecal MOG-IgG production is associated with refractory phenotypes such as ADEM, myelitis, and cortical encephalitis [85,86,87,109], which can be observed from the onset of the disease and is associated with disability and higher CNS inflammation.Complement activation in the blood and CSF [27,93,110], of which CSF C5b-9 is a common biomarker of severity predicting an EDSS > 3.0 [27].Serum neurofilament L, a marker of neural damage, correlates with clinical attacks of nadir [90,111], EDSS [105], recent relapses within 60 days, seizures [89], brain MRI lesions such as ADEM [89,112], and longitudinal spinal cord lesions [112].Glial fibrillary acidic protein (GFAP) does not increase in the serum [105] or CSF [113]; however, its increase in the serum is associated with EDSS [114].

Altogether, these indicators support a stratified treatment approach that allows for early escalation to second-line agents such as rituximab or tocilizumab in high-risk patients. It is important to analyze biomarkers related to various benign and malignant factors that may influence clinical outcomes, including disease relapse activity and worsening of disability (Table 1).

## 5. MOGAD Treatment

### 5.1. Acute-Phase Treatments

The cornerstone of acute-phase MOGAD remains high-dose intravenous methylprednisolone (IVMP), typically administered at 1000 mg/day for 3–5 days, followed by a gradual oral steroid taper over weeks to months [42]. This approach often effectively suppresses acute inflammation and induces neurological recovery, particularly during initial episodes. However, emerging data suggest that a subset of patients is relatively unresponsive to corticosteroid therapy, failing to achieve functional recovery, or experiencing relapse during tapering or shortly after withdrawal [50]. Steroid unresponsiveness has been reported in 20–30% of patients, depending on the cohort and disease phenotype [51]. These observations underscore the need for timely treatment and alternative strategies in poor responders.

### 5.2. Adjunct Therapies: PLEX and IVIG

#### 5.2.1. PLEX

PLEX has demonstrated efficacy in rapidly reducing circulating autoantibodies and improving neurological outcomes in patients with insufficient corticosteroids. It is particularly beneficial in patients with severe ON, TM [81], or fulminant encephalitis [43]. PLEX is typically initiated within 7–10 days of steroid failure and involves 5–7 sessions. In addition, it is suggested that first-line apheresis (within 2 days) and concomitant use of disease-modifying therapy, such as rituximab, azathioprine, and mycophenolate mofetil in addition to glucocorticoid use, are associated with complete remission in 21%, partial remission in 70%, and no remission in 9%, and associated with favorable EDSS improvement [83].

#### 5.2.2. IVIG

As an adjunct or monotherapy, IVIG has gained attention because of its potential superiority in certain refractory or pediatric-onset cases. Several studies, including randomized pediatric trials, have shown that IVIG may lead to faster functional recovery and reduced relapse rates than steroids alone [115,116,117]. Its mechanisms likely include Fc receptor blockade, neutralization of inflammatory cytokines, and modulation of autoreactive B and T cells, wherein patients relapse during steroid tapering. Moreover, monthly IVIG has shown promise in preventing attacks and reducing steroid exposure [118].

### 5.3. Maintenance Therapy

#### 5.3.1. Conventional Treatments

Azathioprine and mycophenolate mofetil: These oral immunosuppressants are widely used in NMOSD and, by extension, have been tested in MOGAD. Observational studies and meta-analyses suggest that both agents can reduce the frequency of relapse in MOGAD, particularly when used early and in combination with oral corticosteroids during initiation [4,5]. However, relapse is common despite treatment, particularly after steroid withdrawal. Furthermore, the delayed onset of efficacy (typically >3 months), potential adverse effects, and the need for regular laboratory monitoring limit its use, particularly in pediatric populations [6].

Rituximab: A monoclonal antibody targeting CD20^+^ B cells is highly effective against AQP4-IgG^+^ NMOSD, but shows inconsistent benefits in MOGAD [119]. Although some patients respond well, others relapse despite complete B cell depletion [120,121]. This paradox may reflect the fact that MOG-IgG production is often extrafollicular or partially T cell-driven, and rituximab does not eliminate long-lived plasma cells or influence T cell–mediated mechanisms as it eliminates B cells that act as antigen presenting cells [37]. Recent cohort studies have noted that rituximab reduced relapse rates by 37%; however, up to 67% of patients relapsed within 2 years of rituximab initiation despite apparent robust B cell depletion, raising concerns about its role as a first-line agent in MOGAD maintenance [119].

IVIG: Increasing evidence supports IVIG as a favorable long-term treatment, particularly for children and patients intolerant or unresponsive to traditional immunosuppressants. Monthly IVIG administration significantly reduces relapse rates in pediatric and adult patients with MOGAD, with a favorable safety profile and fewer systemic adverse effects [115,116,117]. Mechanistically, IVIG exerts pleiotropic immunomodulatory effects, including Fc receptor blockade, pathogenic autoantibody neutralization, interference with the effector functions of B and T cells, and cytokine network rebalancing, which may explain its efficacy in MOGAD [122].

#### 5.3.2. Emerging Therapies in Clinical Trials and Others

However, a subset of patients remains refractory to conventional immunotherapy. To address this issue, novel therapeutic strategies targeting diverse immune pathways, including inhibitors of interleukin-6 (IL-6) receptors, neonatal Fc receptors (FcRn), and the complement cascade, and pan-B cell depletion therapies, are being investigated.

#### 5.3.3. IL-6 Receptor Inhibitors: Tocilizumab and Satralizumab

IL-6 plays a central role in B cell differentiation, T cell activation, and pro-inflammatory signaling. Case series have reported favorable outcomes with tocilizumab, an IL-6 receptor blocker, in patients with highly relapsing MOGAD unresponsive to steroids, rituximab, or IVIG [58,72,92]. These patients often experience a marked reduction in the relapse rate and stabilization of neurological function. More recently, satralizumab, a long-acting subcutaneous IL-6R inhibitor approved for the treatment of AQP4^+^ NMOSD, was evaluated in MOGAD cohorts. Now a phase 3 trial of satralizumab (WN43194) is currently enrolling patients.

#### 5.3.4. FcRn Antagonists: Rozanolixizumab and Efgartigimod

FcRn inhibitors block IgG recycling, leading to the rapid and sustained depletion of pathogenic autoantibodies, including MOG-IgG. Rozanolixizumab and efgartigimod have shown promising results in the treatment of myasthenia gravis and other IgG-mediated diseases. Because MOGAD is antibody-mediated and frequently relapses, blocking FcRn is a rational therapeutic strategy. A phase 2 trial of rozanolixizumab (NCT05835420) enrolling patients with MOGAD and preliminary experimental results have suggested a rapid decline in serum MOG-IgG titers [123,124]. Now a phase 3 trial of rozanolixizumab (MOG001) is currently on-going.

#### 5.3.5. Complement Inhibition

Although complement activation is less prominent in MOGAD than in AQP4^+^ NMOSD, necrotizing lesions and C9neo deposition have been documented in severe MOGAD pathology [7,8], suggesting a possible role in a subset of refractory or fulminant cases. Currently, the use of the anti-C5 antibodies eculizumab and ravulizumab is limited in AQP4^+^ NMOSD, but has been used in non-AQP4+NMOSD, which shows promising treatment effects [125]. In a Swiss nationwide survey, the complement inhibitor anti-C5 eculizumab was suggested as second-line therapy after rituximab or as third-line therapy after second-line anti-IL-6 treatments [126]. Several candidates for the upstream anticomplement mechanisms in MOGAD warrant clinical trials [127].

#### 5.3.6. B Cell Depletion Therapy

In parallel, B cell-targeting agents, such as ocrelizumab (anti-CD20) and inebilizumab (anti-CD19), are under consideration. CD20-depleting therapy has yielded mixed results in MOGAD, potentially owing to the persistence of CD20^−^ plasma cells. In contrast, anti-CD19 agents target a broad range of B lineage cells, including plasmablasts, and are being explored for multiphasic or treatment-resistant MOGAD, although evidence remains limited [128].

#### 5.3.7. Chimeric Antigen Receptor T Cell Therapy and Autologous Hematopoietic Stem Cell Transplantation (aHSCT)

A particularly aggressive case has been reported in response to aHSCT, highlighting the potential role of aHSCT in select patients [73]. Most notably, CD19-targeted CAR T-cell therapy has been applied to treat refractory MOGAD, showing sustained suppression of relapse activity and immunological response for over 1 year [129,130]. Although MOGAD is often considered a benign or monophasic illness, a subset of patients, particularly those with bilateral ON, cortical encephalitis, or conus-involving myelitis, requires intensive immunosuppression and long-term management. Identifying early clinical and imaging markers of disease severity is critical for optimizing outcomes.

## 6. Discussion: Toward a Clinical Algorithm for Difficult-to-Treat MOGAD

Although most patients with MOGAD respond to standard immunotherapy, a distinct subset exhibits primary resistance or highly relapsing disease. For these patients, a structured clinical algorithm, which was reviewed in the European consensus meeting on pediatric MOGAD cases, is useful to guide timely diagnosis, treatment escalation, and individualized care [82].

### 6.1. Proposed Escalation Pathway: Acute Phase

A pragmatic treatment algorithm for refractory MOGAD should begin with:First-line acute therapy: IV methylprednisolone (1 g/day × 3–5 days, 1–2 sessions), followed by oral glucocorticoid treatment, initiated usually from 0.5~1.0 mg/kg, tapering over ≥3 months, followed by slow tapering from 10 mg.If the response is incomplete: PLEX (5–7 sessions) and IVIG (2 g/kg for 5 days).If the response is incomplete: rituximab or cyclophosphamide (1 g, monthly).

### 6.2. Proposed Escalation Pathway: Maintenance Therapy

Persistent or early relapse: initiation of maintenance immunotherapy, often beginning with monthly IVIG, oral azathioprine/MMF/tacrolimus, or rituximab.Second-line biologics: tocilizumab and satralizumab after trying rituximab for patients with frequent relapses.Other options depend on the local situation; in severe multiphasic cases, options include anti-complement treatment (eculizumab, ravulizumab), aHSCT or anti-CD19 CAR T cells under specialized care.

### 6.3. Tailoring Therapy to Phenotype and Biomarkers

Personalized medicine is becoming increasingly viable for MOGAD owing to growing insights into various biological and radiological phenotypes, immune markers, and therapeutic response patterns. Persistent MOG-IgG seropositivity or escalated MOG-IgG titration predicts relapse activity, increases in immunosuppressant dose, or the initiation of biologics [47]. Intrathecal MOG-IgG detection predicts refractory phenotypes of MOGAD derived from limited biological efficacy [86]. Elevated IL-6 or complement C5b-9 levels that predict disease activity and worsening of disability may support escalation to second-line biologics [73,129]. The cortical encephalitis phenotype requires more aggressive, prolonged immunotherapy and seizure treatment, and requires electroencephalography and MRI during clinical worsening [44]. Leukodystrophy-like presentations require early immunotherapy with biological treatments and neurodevelopmental follow-up, particularly in pediatric cases [65].

## 7. Future Directions and Research Priorities

Despite the growing recognition of MOGAD as a distinct CNS autoimmune demyelinating condition, several clinical questions remain unanswered, particularly regarding the treatment-refractory or severe phenotypes. Coordinated international efforts are crucial to advance this field through improved data sharing, biomarker discovery, and therapeutic innovation. Currently, data on severe or refractory MOGAD are fragmented across case reports, small series, and retrospective cohorts. Establishing international registries is critical for designing prospective trials and establishing evidence-based guidelines. Most therapeutic data on MOGAD have been derived from uncontrolled retrospective studies, and randomized controlled trials (RCTs) specifically targeting high-risk populations are required. In addition to IL-6R inhibitor and anti-FcRn trials, RCTs evaluating agents such as B cell depletion, cell-based CAR-T therapies, and anti-complement treatments in these populations will be critical for refining current algorithms [129].

## 8. Conclusions

To improve outcomes in patients with severe or refractory MOGAD, future efforts must combine clinical standardization through patient registries, rigorous clinical trials, and translational research on immunological and imaging biomarkers. Precision medicine frameworks and prospective data capture hold promise for transforming MOGAD care.

## Figures and Tables

**Figure 1 ijms-26-08538-f001:**
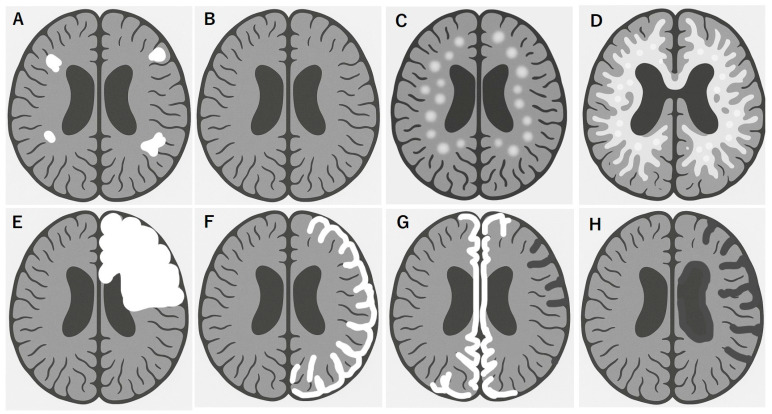
Brain MRI features of MOGAD link to disease worsening and poor recovery. Multifocal white matter lesions, shown in white (**A**), are commonly observed in MOGAD, especially in the deep white matter and corticomedullary junction, in addition to optic and spinal cord lesions, which is resolved in typical benign form of MOGAD (**B**). In contrast, some ADEM-like aggressive forms of MOGAD tend to rapidly grow multiple white matter lesions (**C**), which results in diffuse transitional lesions like Schilder’s sclerosis (**D**). In other cases, solitary or multifocal tumefactive disease is observed (**E**). Another MOGAD brain lesion is cortical encephalitis with or without meningeal enhancement, typically located in a hemispheric unilateral pattern (**F**) or medial bilateral pattern (**G**) of lesions, which should be carefully checked with focal signs of hemiparesis or paraparesis in addition to focal epileptic seizures. Some of these lesions may have a clinically refractory course to immunotherapy and anti-epileptic treatments, preventing a healthy development, especially in school children, and resulting in focal brain atrophy (**G**) and hemispheric brain atrophy (Rasmussen-type) (**H**).

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
