# Peer review of "Myelin Oligodendrocyte Glycoprotein Antibody-Associated Disease: Pathophysiology, Clinical Patterns, and Therapeutic Challenges of Intractable and Severe Forms"

_ijms, 2025, doi:10.3390/ijms26178538_

Round 1

Reviewer 1 Report

Comments and Suggestions for Authors

Τhe study Myelin Oligodendrocyte Glycoprotein Antibody-Associated Disease (MOGAD): Pathophysiology, Clinical Patterns, and Therapeutic Challenges of Intractable and Severe Forms is a well written review on mogad pathophysiology and also summarizes previous and current knowledge on treatment stradegies.

Major revisions

  • Table 1 needs some references in a separate row, also include intrathecal MOG or CSF MOG positivity.
  • Figure 1 needs explanation of A to H and could benefit the addition of optic nerve image, if possible
  • Comment more on the topic. Should MOGAD and NMOSD continue to be regarded as purely relapse-associated, or is there a spectrum of low-grade progression that warrants further study? The review does not sufficiently address whether neuroaxonal injury or neurodegeneration may occur independently of active demyelination or inflammation. An addition to this topic would be evidence on cognitive function in MOGAD.
  • Authors should try to separate if there are biomarkers of either severe attack or relapsing disease. Thus, studies that stratify biomarker profiles by disease phase (monophasic vs. multiphasic) and severity (mild vs. fulminant) are urgently needed.
  • Mog antibody pathogeneicity should be enhanced by some animal models like eg in a macaque model in which complement dependent vacuolisation of myelin, lesion pathology, EAE and MOG Ab were observed after injection of recombinant human MOG

Minor

Line 35; Mogad manifest as diverse phenotype rephrase and write MOGAD patients manifest diverse clinical phenotypes

Line 48: AQP4 seronegative NMOSD and not AQP4 IgG- adults

Line 69:  these histopathological….a reference is needed, and author should give some examples of patients in literature

Line 91: the three pathways…I can see two pathways. Make more clear.

Line 81 mainly IgG1 can mediate….please add an “, and have been shown to mediate…” add reference

Line 99; MOGAD is more commonly associated with pattern II. Pattern III is associated with AQP4. Please provide more comprehensive literature is the immunopatterns.

LINE 110;  rephrase as there is no explanation regarding change in cytoskeleton eg adcc results in alterations in cytoskeleton and formation of thin filaments…

Line 127; hyper simplified sentence…is not a mild disease…. Rephrase and give some references.

Line 154-156; Please compare findings with MS and NMOSD with regard to OCT findings and rephrase the sentence. Please specific if the findings are is acute or chronic forms of disease.

Line 271; rephrase . I don’t understand the redundant ….of which milder …as a bad sign

Line 336 rituximab influences T cells mediated mechanisms as it eliminates B cells that act as antigen presenting cells in T cells, so please rephrase

Give abbreviations for  aaHSCT, CAR T-cell when introducing the words

Have the some grammar size and type in the article

Comments on the Quality of English Language

Needs improvement

Author Response

Reviewer 1:

Τhe study Myelin Oligodendrocyte Glycoprotein Antibody-Associated Disease (MOGAD): Pathophysiology, Clinical Patterns, and Therapeutic Challenges of Intractable and Severe Forms is a well written review on mogad pathophysiology and also summarizes previous and current knowledge on treatment stradegies.

Major revisions

  • Table 1 needs some references in a separate row, also include intrathecal MOG or CSF MOG positivity.

Thank you so much for your comment. I add references in a Table according to your comment up to two citations in each factor.

  • Figure 1 needs explanation of A to H and could benefit the addition of optic nerve image, if possible

Thank you so much for this comment. I add a figure legend. It is better to add optic nerve, but many studies suggested optic neuritis is a good prognostic factor (it is incorrect, but not the best prognostic one), so I mention this point in detail). In addition, my focus is to clarify the higher risk of ADEM, multiphasic ADEM, and cortical encephalitis for most severely types including diffuse sclerosis (Schilder-type) and hemispheric atrophy (Rasmussen-type), so I would like not to add optic nerve in this figure and add some sentences in abstract and discussion about this topic.

  • Comment more on the topic. Should MOGAD and NMOSD continue to be regarded as purely relapse-associated, or is there a spectrum of low-grade progression that warrants further study? The review does not sufficiently address whether neuroaxonal injury or neurodegeneration may occur independently of active demyelination or inflammation. An addition to this topic would be evidence on cognitive function in MOGAD.
  • Thank you so much for this very useful comments. Previous paper focus on this optic is very limited, but I would like to add a paragraph of cognitive impairment and mention about this topic as followed:
  • Cognitive impairment
  • Cognitive impairment in MOGAD is mostly a sequela of disease onset or relapsed symptoms, but it might be underestimated a progressive deterioration of cognition in MOGAD as observed in smoldering MS lesions[70]. In an adult cohort including 32 MOGAD cases (median age 29.4 years), several types of cognitive impairments were observed, such as mental flexibility (16.7%), attention (11.1~14.8%), and verbal working memory (10.3%) with reduced volumes of cerebral white matter and gray matter compared with controls, which is associated with a history of ADEM/ADEM-like episodes [71]. In contrast, in a pediatric MOGAD cohort (n=109), in a median follow-up of 1.6 years, 15 of 82 cases and 14 of 59 cases with brain lesions had learning difficulties, probably because of reduced brain growth observed in 86 of 109 (79%) patients in acute stage, and 50 of 109 (46%) had brain atrophy at follow-up driven progressively by brain lesions [72].

  • Authors should try to separate if there are biomarkers of either severe attack or relapsing disease. Thus, studies that stratify biomarker profiles by disease phase (monophasic vs. multiphasic) and severity (mild vs. fulminant) are urgently needed.

Thank you so much for this very important suggestion. I agree with you the importance of several factors related to “Relapse” and “Disability”, which should not be confused. However, it is also hard to completely separate these papers, because of several factors related to both factors. So I tried to add “R”elapse and “D”isability marks in each factor in Table 1 with related references, which I hope it would be helpful for readers. Moreover, I add real data related to relapse and disability in table 1 as much as possible for readers to easily imaging original contents.

  • Mog antibody pathogeneicity should be enhanced by some animal models like eg in a macaque model in which complement dependent vacuolisation of myelin, lesion pathology, EAE and MOG Ab were observed after injection of recombinant human MOG

Cellular immunity, including MOG-specific an

According to your comment, I would like to add a new paragraph ”Cellular immunity, including MOG-specific and innate immunity” , which is focusing on existed ADCP and other MOG-specific autoimmunity including EAE models.d innate immunity

Other aspect of MOG-IgG cytotoxicity is antibody-dependent cellular phagocytosis (ADCP), which could be identified by its functional role on in vitro MOG-expressing cells [21]. MOG-IgG includes not only IgG1 but also IgG2, IgG3, and IgG4, all of which could induce ADCP;, in contrast, CDC could be observed by IgG1 and IgG3, suggesting that patient-s’ derived antibodies must have multiple mechanisms of cytotoxic effects against targeted myelin sheaths [21], possibly linkeds to a unique accumu-lation of macrophages in perivenous demyelinating lesions in the ADEM phenotype of MOGAD [8]. In EAE, MOG-specific CD4+ and CD8+ T cells are required for the induction of EAE in B -cell- independent conditions, which can explain the suboptimal re-sponses toof anti-CD20 therapy in MOGAD   patients with MOGAD compared with NMOSD [39]. Several epitopes of MOG-specific T cells have beenwere reported in patients with MOGAD, patients including extracellular and intracellular epitopes [40, 41], consistent with various epitopes of MOG-IgG [32], suggesting these various powers for encephalitogenic roles and possible epitope spreading. Pathogenic patient-derived MOG-IgG enhanced cognate MOG-specific T cells and macrophages infiltrations in experimental models, which could drive the tissue inflammation and demyelination via an ensemble of various T and B cells epitope -spreading and disease worsening [32]. CFor considering itsthe longitudinal history, there is no doubt that MOG-associated disease has been most deeply studied regardingabout several points of the above-mentioned pathological mechanisms, some of which must suggest upstreaming factors related to thedisease worsening of MOGAD.

Minor

Line 35; Mogad manifest as diverse phenotype rephrase and write MOGAD patients manifest diverse clinical phenotypes

Thank you so much. I add “clinical” in this sentence.

Line 48: AQP4 seronegative NMOSD and not AQP4 IgG- adults

 Thank you so much. I add “’in children with ADEM/encephalitis and adults with AQP4-IgG seronegative NMOSD” in this sentence.

Line 69:  these histopathological….a reference is needed, and author should give some examples of patients in literature

Thank you so much. I add a sentence “clinically relevant T2-lesions in MOGAD resolve more completely and frequently than those in MS and NMOSD [19]”. I add a citation in this point.

Line 91: the three pathways…I can see two pathways. Make more clear.

Thank you so much. I addwhich are activated when mannose-binding lectin encounters conserved carbohydrate motifs in pathogens (Lectin pathway)” in the sentence.

Line 81 mainly IgG1 can mediate….please add an “, and have been shown to mediate…” add reference

Thank you so much. I add citations in this sentence.  

Line 99; MOGAD is more commonly associated with pattern II. Pattern III is associated with AQP4. Please provide more comprehensive literature is the immunopatterns.

Thank you so much. I omit about the pattern III for better understanding the complicated mechanisms in MOGAD.

LINE 110;  rephrase as there is no explanation regarding change in cytoskeleton eg adcc results in alterations in cytoskeleton and formation of thin filaments…

Thank you so much. I revise this sentence as followed: In an in vitro study, ADCC induces the striking loss of the thin filaments and microtubule cytoskeleton in cultured oligodendrocytes [36]. In an in vivo microinjection model, MOG-IgG itself without complement can cause myelin changes and altered expression of axonal proteins without marked inflammation and other tissue damage and could recover well within 2 weeks [37]. In addiion, in an EAE model, MOG-IgG-induced demyelination is equally mediated by CDC and ADCC, suggesting a diverse mechanism of pathological characteristics involved in MOGAD, with or without complement-induced demyelination and tissue necrosis [38]. Moreover, its ADCC can be activated by FcγR activation [38], which may also be essential for cognate T cell activation via antigen-presenting cells.

Line 127; hyper simplified sentence…is not a mild disease…. Rephrase and give some references.

Thank you so much. I revise this point as “even though it is not a mild disease [44]”.  

Line 154-156; Please compare findings with MS and NMOSD with regard to OCT findings and rephrase the sentence. Please specific if the findings are is acute or chronic forms of disease.

Thank you so much. I add “Optical coherence tomography (OCT) has suggested that peripapillary retinal nerve fiber layer (pRNFL) thickness measured acutely frequently demonstrates swelling, and its thickness in acute optic neuritis can differentiate MOGAD from MS [61]. in this sentence.

Line 271; rephrase . I don’t understand the redundant ….of which milder …as a bad sign

  • Thank you so much. I rephrase it as “Bilateral redundant swollen optic nerves in the acute stage are often observed with relatively severe visual acuity loss in the nadir [42, 104], of which a lack of or milder enhancement is a sign for poor prognosis [60]

Line 336 rituximab influences T cells mediated mechanisms as it eliminates B cells that act as antigen presenting cells in T cells, so please rephrase

Thank you so much. I rephrase it as you suggested.

Give abbreviations for  aaHSCT, CAR T-cell when introducing the words

Have the some grammar size and type in the article

Thank you so much. I revise them as you suggested.

Reviewer 2 Report

Comments and Suggestions for Authors

The author has produced a comprehensive narrative review of myelin-oligodendrocyte-glycoprotein antibody-associated disease (MOGAD), focusing on severe and refractory phenotypes. The manuscript is generally well written and will be useful for clinicians and researchers. Nevertheless, several points need clarification or expansion.

  1. Definitions of “severe” and “refractory” MOGAD  
  • Please provide explicit, operational definitions. For severity, do the authors use an EDSS threshold (e.g., EDSS ≥ 4.0 or ≥ 6.0) or another validated score?  
  • For “refractory disease,” the text mentions “poor functional recovery” without specifying the scale or cut-off. Clarify whether this refers to failure to reach EDSS ≤ 2.0 after three months, absence of ≥ 1-point EDSS improvement after two lines of therapy, or another benchmark. Distinguish clearly between “refractory” and “severe and refractory.”

  1. Pathological insights  
  • MOG-specific T-cell responses: Recent evidence (PMID 38996203) indicates that cytotoxic CD4⁺ and CD8⁺ MOG-specific T cells persist despite B-cell depletion and may explain suboptimal responses to anti-CD20 therapy. Please incorporate this finding.  
  • Antibody affinity and epitope spreading: Discuss how high-affinity MOG-IgG and intramolecular/intermolecular epitope spreading correlate with severe phenotypes (PMID 30014603).

  1. Table 1 – Biomarkers  

For each proposed biomarker, add quantitative data when available:  

  • association with disease severity (correlation coefficients or ORs),  
  • prognostic cut-offs,  
  • sensitivity, specificity, and AUC from validation cohorts.  

If such data are lacking, state this explicitly.

  1. Clinical algorithm  

A flowchart summarizing the stepwise diagnostic work-up and treatment escalation for severe/refractory MOGAD would greatly enhance clarity and practical utility.

Author Response

Reviewer 2:

The author has produced a comprehensive narrative review of myelin-oligodendrocyte-glycoprotein antibody-associated disease (MOGAD), focusing on severe and refractory phenotypes. The manuscript is generally well written and will be useful for clinicians and researchers. Nevertheless, several points need clarification or expansion.

  1. Definitions of “severe” and “refractory” MOGAD  
  • Please provide explicit, operational definitions. For severity, do the authors use an EDSS threshold (e.g., EDSS ≥ 4.0 or ≥ 6.0) or another validated score?  

Thank you so much for this important comment. According to heterogenous clinical features, I would like to add these scores originally from each reference in Table 1, which would be helpful for readers. In addition, some references used modified Ranking Scale of benign case whose score is less than 1 in contrast severer cases had more than 2.

  • For “refractory disease,” the text mentions “poor functional recovery” without specifying the scale or cut-off. Clarify whether this refers to failure to reach EDSS ≤ 2.0 after three months, absence of ≥ 1-point EDSS improvement after two lines of therapy, or another benchmark. Distinguish clearly between “refractory” and “severe and refractory.”

Thank you so much for this important comment. In various factors influenced on this topic, it is not easy to identify them, but I would like to add several useful score changes in each reference including EDSS 1.0 point improvement in maintenance treatment (IVIG) or ARR change as much as possible, which would be useful for readers. For refractory, it is important to show the recovery situation, which could be supported by some references cited in a new version of the manuscript.

  1. Pathological insights  
  • MOG-specific T-cell responses: Recent evidence (PMID 38996203) indicates that cytotoxic CD4⁺ and CD8⁺ MOG-specific T cells persist despite B-cell depletion and may explain suboptimal responses to anti-CD20 therapy. Please incorporate this finding.  
  • Antibody affinity and epitope spreading: Discuss how high-affinity MOG-IgG and intramolecular/intermolecular epitope spreading correlate with severe phenotypes (PMID 30014603).

Thank you so much for your important and valuable comments above mentioned. I would like to add a sentence as followed:

In EAE, MOG-specific CD4+ and CD8+ T cells are required for the induction of EAE in B cell-independent conditions, which can explain the suboptimal responses to anti-CD20 therapy in patients with MOGAD compared with NMOSD [39]. Several epitopes of MOG-specific T cells have been reported in patients with MOGAD, including extracellular and intracellular epitopes [40, 41], consistent with various epitopes of MOG-IgG [32], suggesting various encephalitogenic roles and possible epitope spreading. Pathogenic patient-derived MOG-IgG enhanced cognate MOG-specific T cell and macrophage infiltration in experimental models, which could drive tissue inflammation and demyelination via an ensemble of various T and B cell epitope spreading and disease worsening [32]. Considering its longitudinal history, there is no doubt that MOG-associated disease has been most deeply studied regarding several points of the above-mentioned pathological mechanisms, some of which suggest upstream factors related to the worsening of MOGAD.

  •  
  1. Table 1 – Biomarkers  

For each proposed biomarker, add quantitative data when available:  

  • association with disease severity (correlation coefficients or ORs),  
  • prognostic cut-offs,  
  • sensitivity, specificity, and AUC from validation cohorts.  

If such data are lacking, state this explicitly.

 Thank you so much for your important and valuable comments. I would like to add some data about percentile of ARR reduction, EDSS recovery point > 1.0., or antibody titration, and I would like to try to add points for odds ratio in Table 1, which would be helpful for readers.

  1. Clinical algorithm  

A flowchart summarizing the stepwise diagnostic work-up and treatment escalation for severe/refractory MOGAD would greatly enhance clarity and practical utility.

Thank you so much for this comment. As you know, it is very important to suggest a clinical algorithm. However, it is still arguing against the regional difference of clinical algorithm and undetermined factors in several treatments in child and adult. My main goal in this paper is not making a clinical algorithm but making a more active promotion of discussion for the development of high efficacy treatment on MOGAD including CAR-T and anti-complement drugs. So, in this point, I would like to cite a previous paper of European pediatric MOG consortium consensus algorithm in this sentence and I would like to add some statements about promotion of new promising treatments in abstract and discussion as followed:

It is promising of currently ongoing investigational antibodies against anti-interleukin-6 receptor and the neonatal Fc receptor. Moreover, due to possible refractory issues such as the intrathecal production of autoantibody and the involvement of complement in the worsening of the lesion, further developments of other mechanisms of action such as chimeric antigen receptor T-cell (CAR-T) and anti-complement therapies are warranted in the future.

Round 2

Reviewer 1 Report

Comments and Suggestions for Authors

Accepted